

# RITAN: rapid integration of term annotation and network resources

Michael T. Zimmermann[1,2,3], Brian Kabat[3], Diane E. Grill[3], Richard B. Kennedy[4] and Gregory A. Poland[4]

[1] Bioinformatics Research and Development Laboratory, Genomic Sciences and Precision Medicine Center, Medical College of Wisconsin, Milwaukee, WI, USA
[2] Clinical and Translational Sciences Institute, Medical College of Wisconsin, Milwaukee, WI, USA
[3] Division of Biomedical Statistics and Informatics, Department of Health Sciences Research, Mayo clinic, Rochester, MN, USA
[4] Mayo Clinic Vaccine Research Group, Mayo Clinic, Rochester, MN, USA

Corresponding author
Gregory A. Poland,
poland.gregory@mayo.edu

## ABSTRACT

**Background:** Identifying the biologic functions of groups of genes identified in high-throughput studies currently requires considerable time and/or bioinformatics experience. This is due in part to each resource housed within separate databases, requiring users to know about them, and integrate across them. Time consuming and often repeated for each study, integrating across resources and merging with data under study is an increasingly common bioinformatics task.

**Methods:** We developed an open-source R software package for assisting researchers in annotating their genesets with functions, pathways, and their interconnectivity across a diversity of network resources.

**Results:** We present rapid integration of term annotation and network resources (RITAN) for the rapid and comprehensive annotation of a list of genes using functional term and pathway resources and their relationships among each other using multiple network biology resources. Currently, and to comply with data redistribution policies, RITAN allows rapid access to 16 term annotations spanning gene ontology, biologic pathways, and immunologic modules, and nine network biology resources, with support for user-supplied resources; we provide recommendations for additional resources and scripts to facilitate their addition to RITAN. Having the resources together in the same system allows users to derive novel combinations. RITAN has a growing set of tools to explore the relationships within resources themselves. These tools allow users to merge resources together such that the merged annotations have a minimal overlap with one another. Because we index both function annotation and network interactions, the combination allows users to expand small groups of genes using links from biologic networks—either by adding all neighboring genes or by identifying genes that efficiently connect among input genes—followed by term enrichment to identify functions. That is, users can start from a core set of genes, identify interacting genes from biologic networks, and then identify the functions to which the expanded list of genes contribute.

**Conclusion:** We believe RITAN fills the important niche of bridging the results of high-throughput experiments with the ever-growing corpus of functional annotations and network biology resources.

**Availability:** Rapid integration of term annotation and network resources is available as an R package at github.com/MTZimmer/RITAN and BioConductor.org.

# INTRODUCTION

High-throughput technologies are enabling systems-level assays for an increasing diversity of applications, which is revealing previously unknown genetic underpinnings of many diseases and phenotypes. It is common for researchers to identify a list of potentially interesting genes, many of which have no known relationship to the condition studied. In this situation, it is challenging to identify the functions these genes contribute to, their precise mechanisms, and the role each gene plays in orchestrating those functions. Further, it is necessary to estimate how many of the prioritized genes may be chance associations due to biologic variability. Many data systems and computational tools have been developed to aid researchers in addressing these critical challenges; however, to better understand how genes interact with each other to achieve a common function, the comprehensive integration of multiple annotation and network biology resources typically remains a data wrangling venture re-invented for each research project.

Many resources have been developed to standardize and codify terms or descriptive annotations for the functions of genes or their encoded products. Ontologies such as gene ontology (GO) (*Ashburner et al., 2000*) aim to describe each gene specifically and comprehensively. Pathway databases such as PID (*Schaefer et al., 2009*), KEGG (*Kanehisa et al., 2012*), and others, gather sets of genes participating in a particular function that has defined initiation points and outcomes and the topology that connects them. A third approach is to identify which genes are co-regulated across conditions (functional modules), as exemplified by Hallmarks in MSigDB (*Subramanian et al., 2005*) or blood transcription modules (*Li et al., 2014*). Resources like DisGeNet index relationships between genes and diseases (*Pinero et al., 2015*). Broadening the concept of a pathway, network biology resources including HPRD (*Prasad, Kandasamy & Pandey, 2009*), CCSB (*Rolland et al., 2014*), and STRING (*Szklarczyk et al., 2011*) seek a more comprehensive view—how genes, proteins, and potentially other molecules broadly interact with one another. Each of these resource types has its own strengths and limitations. Therefore, in order to gain the most comprehensive understanding of how a group of genes relate to each other, and what biologic functions they may define or contribute to, data integration is necessary.

We have generated an open-source and publically available software package, RITAN, for the Rapid Integration of Term Annotation and Network resources. For term annotation or enrichment analysis, our system currently indexes 16 resources spanning GO, pathways, and immunologic modules, and facilitates annotation of a list of genes (e.g., those differentially expressed). Users may query one resource at a time or multiple resources together, allowing false discovery to be accounted for across all resources used. For network biology, nine resources are indexed and confidence-score filtering is enabled when available. See our vignettes within the package for the full list of resources. Users may add resources using standard file formats to extend the depth and breadth

of coverage. By using these resources and adding additional resources of interest, analyses can be more efficient, standardized, and reproducible by minimizing manual steps and keeping resources indexed in a common and accessible system. We have used this integrated network resource in our previous studies (*Haralambieva et al., 2016a*, *2016b*; *Kennedy et al., 2016*; *Zimmermann et al., 2016*) to provide an organizational framework to unify results from multiple omics data (mRNA, protein, DNA methylation), yielding an integrated signature that is more informative than each alone.

## METHODS

Rapid integration of term annotation and network resource (RITAN) is implemented as a package in the R programing language (*R Core Team, 2014*). It leverages multiple existing packages, extending their utility, including igraph (*Csardi & Nepusz, 2006*) and STRINGdb. Enrichment analysis currently uses the hypergeometric test.

Reproducible analysis examples are available in package vignettes, which are also available in our Supplemental Data. We also downloaded normalized breast invasive carcinoma mRNA-Seq gene expression data from the Cancer Genome Atlas (TCGA) pan-cancer analysis (*Ding et al., 2018*) for demonstrating selected utilities. We used EdgeR (*McCarthy, Chen & Smyth, 2012*) to calculate differential gene expression. We used annotations for known protein complexes using CORUM, (*Ruepp et al., 2010*) rare diseases using Orphanet, (*INSERM, 1997*) and regulatory programs using NetPath (*Kandasamy et al., 2010*).

We have developed a self-contained executable R-Shiny application to bring some of RITAN's functionality to a broader audience. The app leverages a combination of technologies including the chromium web browser, electron stand-alone server, and RPortable, and built into a self-contained Windows executable using node.js.

## RESULTS

Rapid integration of term annotation and network resource is an enabling tool for identifying the functional associations within datasets generated by high-throughput methods. We have implemented features that help researchers to uncover relationships within their dataset and augmented by closely related genes in biologic networks (Figs. 1A and 1B). We used a study of anti-TREM1 effects on influenza response, as an example (Figs. 1C and 1D). Data for this study is publicly posted on GEO under the accession ID, GSE9988 (*Dower et al., 2008*). The study considered multiple treatments and lists of genes associated with each comparison of those treatments were published. Using RITAN, multiple resources were combined into a single enrichment analysis, allowing more comprehensive false-discovery control. The visualization made by RITAN shows a study design matrix with enrichment heatmap to visually annotate how different treatments activate different responses (Fig. 1C). Differential pathway activity by each treatment is presented visually and numerically. Genes associating with up-regulated responses are further contextualized by identification of their known molecular inter-relationships (Fig. 1D). The network information can be directly mined, used in computation, or visualized. In this case, we have uploaded the integrated network returned by RITAN to Cytoscape for visualization and scaled each gene by the number

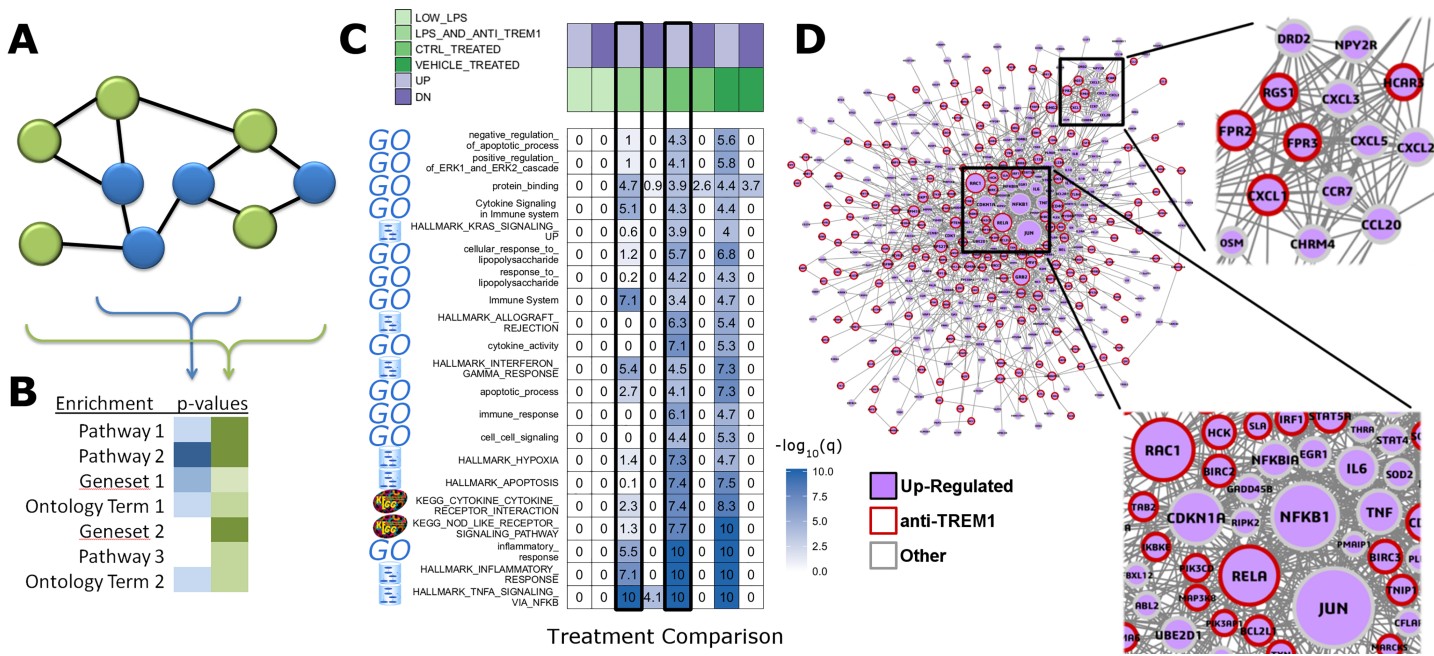

**Figure 1 RITAN facilitates rapid and comprehensive annotation and network integration.** We use the arbitrarily chosen example of GSE9988, a study of anti-TREM1 effects on influenza response. This study considered multiple treatments and lists of genes associated with each comparison of those treatments have been published. (A) From an input gene list (blue nodes), RITAN leverages multiple network resources to identify neighbors (green nodes) and (B) performs integrated term enrichment. We use GSE9988 (*Dower et al., 2008*), a study of anti-TREM1 effects on influenza response, as an example. Using RITAN, multiple resources were combined into a single enrichment analysis with results presented as a heatmap where greater intensity indicates greater statistical significance. The values in the heatmap are the −log$_{10}$($q$-value) with values capped at 10 for visualization. To better use the color scale. Adding a study design matrix above the enrichment heatmap (also by RITAN) visually reveals that significant enrichment scores are defined by the up-regulated gene lists and that they capture aspects of inflammation, innate immunity, cytokine signaling, and interferon signaling. Considering the two comparisons emphasized by a black border, LPS and anti-TREM1-antibody vs. control-treated, there are many terms and pathways exhibiting significant differences in enrichment. (C) RITAN also integrates across multiple network biology resources. The genes associating with up-regulated responses in Fig. 1B are further contextualized by identification of their known inter-relationships. The network information can be directly mined, used in computation, or visualized. In this case, we have uploaded the integrated network returned by RITAN to Cytoscape for visualization. We have scaled each gene by the number of interactions they have as an indication of how "central" they are to response. The network facilitates hypothesis generation by visually displaying the differential association with anti-TREM1 response (indicated by red outline) with different NFkB subunits, differential Akt regulation via JUN, PTEN, GRB2, etc., and different neutrophil chemotactic signals (e.g., FPR3 or CXCL3). Thus, the network is a hypothesis generating tool, focused by first identifying the most relevant subset of results from our integrated enrichment analysis. (D) RITAN integrates network biology resources, which can be directly visualized or imported into Cytoscape. In this example, JUN, NFKB1, TNF, and multiple cytokines were not differentially expressed, but their interactions with the differentially expressed genes were identified, implicating their activity in cellular responses.

of interactions they have as an indication of how "central" they are to response. The network facilitates hypothesis generation by visually displaying the differential association with anti-TREM1 response with different NFkB subunits, differential Akt regulation via JUN, PTEN, GRB2, etc., and different neutrophil chemotactic signals (e.g., FPR3 or CXCL3). Thus, the network is a hypothesis generating tool, focused by first identifying the most relevant subset of results from our integrated enrichment analysis. Through data within biologic networks, genes that were not differentially expressed, but closely interact with differentially expressed genes, can be identified, implicating their potential activity change despite no mRNA expression change.

In addition to facilitating rapid functional annotation, RITAN includes tools to analyze the relationships between resources. For example, many diseases are contributed to by

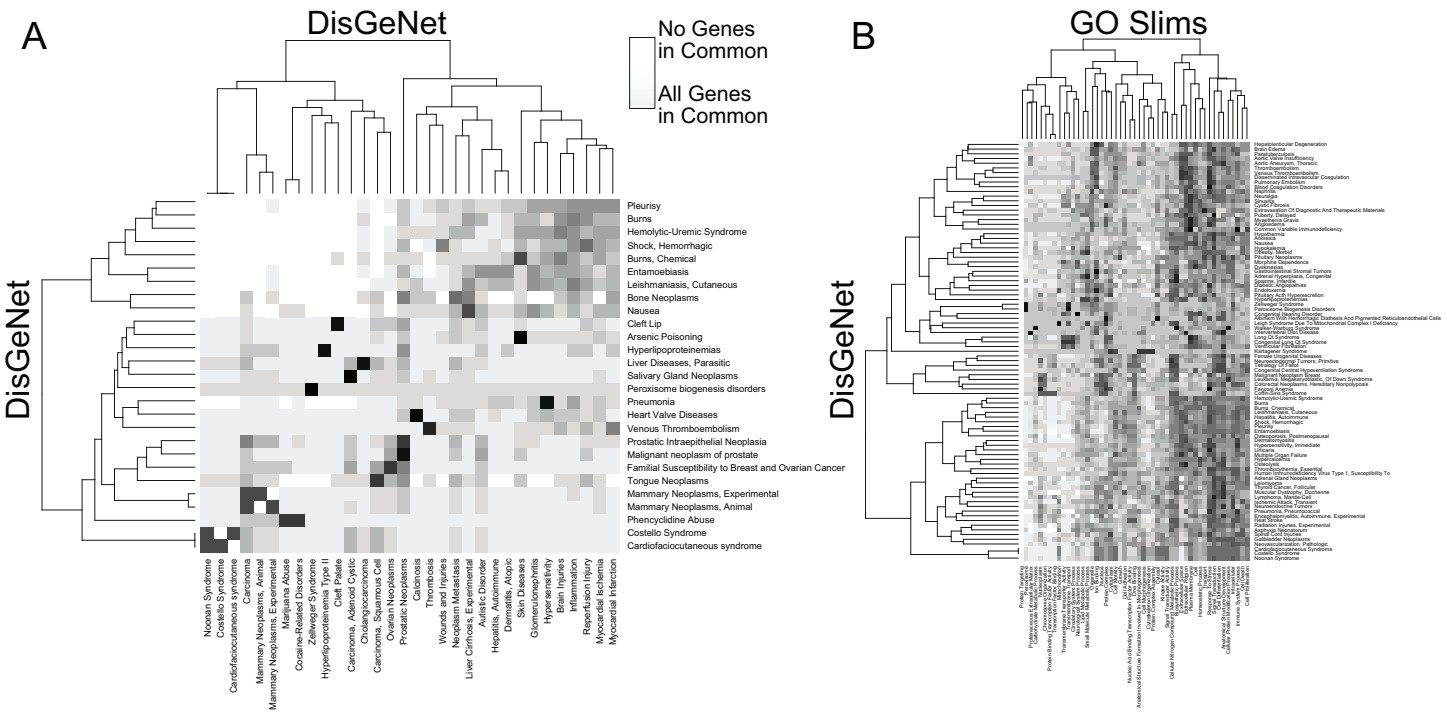

**Figure 2 RITAN facilitates comparing resource similarities.** (A) Within the DisGeNet resource, we compare the overlap of genes annotated to each disease. A disease is shown if it shares at least an 80% overlap with another disease, ignoring self-overlap. (B) Between two resources, DisGeNet, and Go Slim, we show interactions among terms when any pair shares at least 95% overlap.

similar sets of genes (Fig. 2A). Additionally, many diseases are characterized by alteration of specific biologic processes or molecular functions (Fig. 2B). By identifying how resources overlap (fraction of shared genes), we are able to combine the resources in a way that reduces the overlap. Once reduced to a set of terms or pathways that are more independent, enrichment analysis can be run on the combined set of terms and using a more accurate false discovery adjustment. Without reducing overlapping terms, false discovery adjustment across resources will be overly conservative. Additionally, having many resources in the same system allows exploration of relationships among them, and making novel combinations.

We use RITAN as a simple knowledge management system that facilitates data annotation and hypothesis exploration—activities that are nor supported by other tools or are challenging to use programmatically. As additional demonstrations of the utility enabled by RITAN, we investigated differential gene expression profiles between breast cancer subtypes from TCGA. This analysis is to demonstrate an approach to data annotation and exploration enabled by RITAN, rather than a definitive analysis of this specific and well-studied dataset. While we distribute many resources with RITAN, other resource licenses prohibit redistribution. We provide recommendations for additional resources (see our GitHub page; we plan to implement an automated download and setup script). Our use of simple and accepted file formats simplifies the process of adding resources. Leveraging additional resources, we annotated the differentially expressed genes between subtypes, first identifying which of them are shared between specific subtypes,

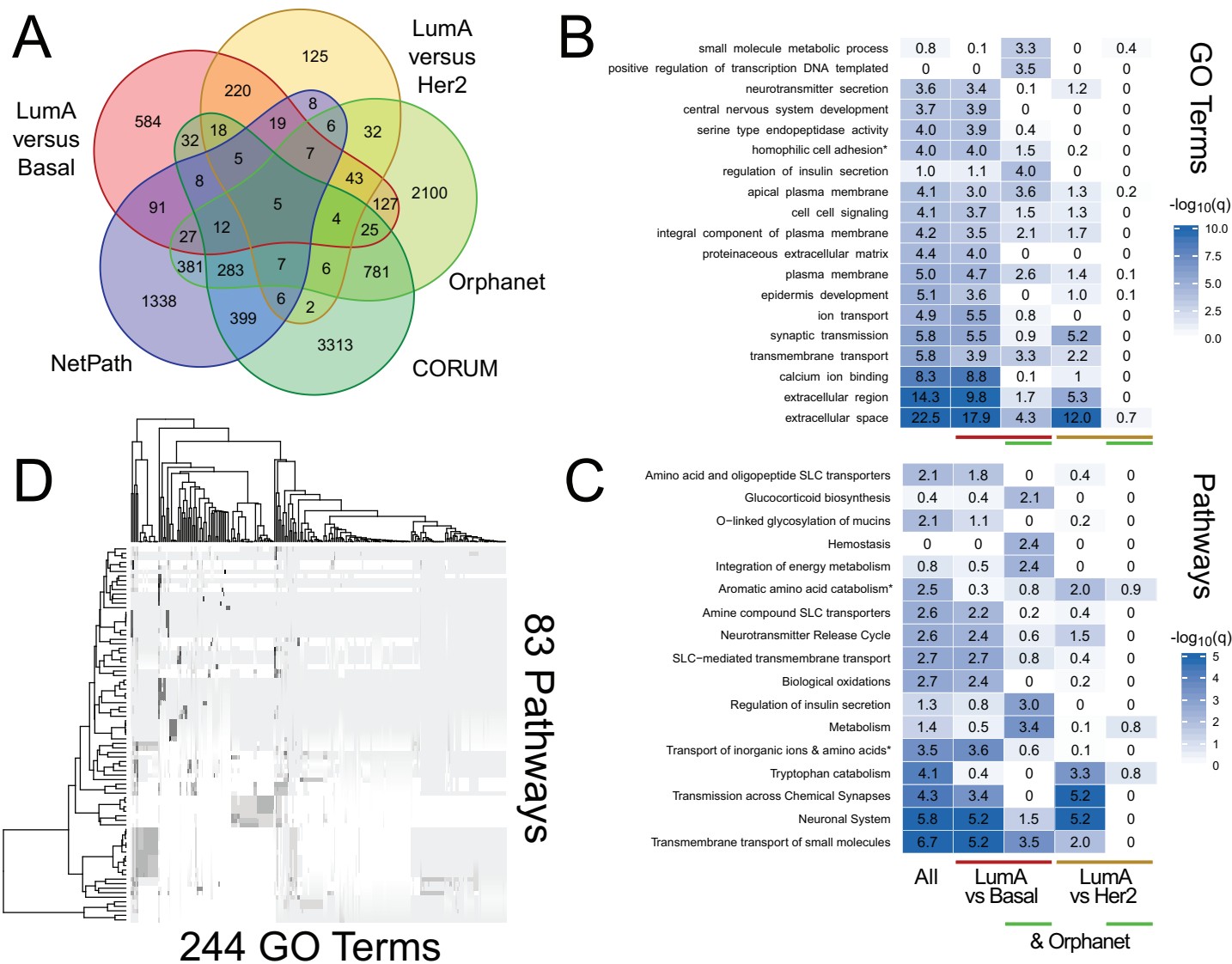

**Figure 3 RITAN facilitates hypothesis generation and exploration.** We used TCGA BRCA differential gene expression data and RITAN to explore specific hypotheses. (A) We annotated differentially expressed genes among luminal A (LumA), Basal, and HER2+ subtypes by their presence in protein complexes, rare disease definitions, and known regulatory programs. The overlapping genes for each combination of annotations indicates additional hypotheses to explore. We used a five-way Venn diagram to show how many genes have each annotation. (B) An example of an additional hypothesis is how functional term enrichment may differ, not only between disease subtypes, but also by contribution to rare disease definitions. We show enrichment plots by subgroup formatted similar to Fig. 1. (C) Additionally, how pathway activation may differ. (D) Similar to Fig. 2, RITAN can be used to explore relationships among resources and to filter each resource to a unique subset. The analysis script for this figure is short (93 lines including comments and plotting; available in Supplemental Data), emphasizing the flexibility and simplicity afforded by RITAN.

part of known protein complexes, define rare diseases, or have clearly defined regulatory programs (Fig. 3A). Multiple protein complexes have differentially expressed genes between disease subtypes including SNARE and the chromosomal passenger complex. Both of these complexes have known roles in carcinogenesis and breast cancer (*Kabisch et al., 2015*; *Meng & Wang, 2015*). Multiple rare disease genes are differentially expressed and contribute to precursor B- and T-cell acute lymphoblastic leukemia, among others.

Next, we considered GO term (Fig. 3B) and Reactome (*Croft et al., 2011*) pathway (Fig. 3C) enrichment among genes differentially expressed between disease subtypes, and for the further subset of differentially expressed genes that contribute to rare diseases. These annotations highlight the role of metabolic differences among disease subtypes and that these functional overlap rare disease mechanisms, complementing the use of rare disease annotations for prioritizing functional somatic variants (*Ma et al., 2017*). Finally, we compared GO terms to Reactome pathways to identify which were shared and which were unique. There are 5,231 GO terms with a one-to-one mapping to 1,607 Reactome pathways (100% of genes shared), and 83 GO terms that highly overlap 244 Reactome pathways (95% of genes shared; Fig. 3D). Further vignettes are available within the package and are included in our Supplemental Data. These analyses among gene annotation resources facilitate hypothesis generation and exploration.

## DISCUSSION

Enrichment analysis has become commonplace in bioinformatics and biostatistics; however, current tools have limitations. First, most resources are distributed independently of each other and integrating across them is left to individual users. Second, each enrichment or pathway analysis is typically performed independently, preventing false discovery adjustment across all tests performed. Further, while web servers are convenient for small numbers of queries, programmatic solutions are an advantage for data analysis workflows and scripting for reproducible research.

Enrichment analysis is a statistical method wherein an input list of genes is compared to a term, function, or pathway definition. The definition is usually a pre-existing list of genes curated to describe a specific cellular function or observation. The number of genes in the pathway is recorded and combined with the size of the input list, the size of the term or pathway, and the total number of genes that were assayed (e.g., all human protein coding genes expressed in the tissue of interest), an overrepresentation statistic is computed. When this statistic is significant (e.g., occurring with a probability less than 0.01 by chance and after false discovery correction), the term is typically ascribed to the input list and we describe the function as "enriched for" within the input list. Thus, defining all three areas is critical: the input list, the term definitions, and the universe of possible gene results.

We encourage users to decide upon an analysis strategy prior to running their analysis, including consideration of which resources are most appropriate for their dataset/experiment, statistical significance thresholds, background geneset, etc. The ease with which RITAN facilitates multi-resource query may lead users to "try one more test," leading to an increased number of hypothesis tests made that may not be accounted for by multiple testing correction. To prevent this and appropriately maintain false-discovery correction, we encourage users to "include one more resource"—to add an additional resource to a single query in RITAN.

Other tools that fill similar roles to RITAN exist, but each within a specific niche, using specific data sources, or having additional input requirements. Within the R framework, GAGE (*Luo et al., 2009*) provides access to GO and KEGG database for multiple species. Reactome offers related services, but annotations are specific to their database.

Web servers for enrichment analysis have been developed including WebGestalt (*Wang et al., 2013*) and DAVID (*Da Huang, Sherman & Lempicki, 2009*). The flexibility of RITAN and ability to integrate and explore relationships among both geneset and network resources, distinguishes RITAN from existing software.

RITAN allows annotation integration across many publically available resources; thus, it facilitates rapid development of novel hypotheses about the potential functions achieved by prioritized genes and multiple-testing correction across all resources used. The annotations from RITAN can be leveraged by existing down-stream analysis tools, such as Cytoscape (*Shannon et al., 2003*) or igraph (*Csardi & Nepusz, 2006*), in order to provide greater visualization and network-analytic power. As illustrated in Fig. 1, the two major features of RITAN are an integrated system for term enrichment and connections among genes via network biology, facilitating reproducible research and hypothesis generation. For analyses across resources, it is important to consider how similar the resources are to each other. Similarities may occur within a pathway resource or even between resource types. For example, within the KEGG pathway resource, JAK/STAT, and MAPK signaling share SOS, RAS, RAF, and other genes. Ontology, pathway, and network information can be applied to multiple entities: gene-centric, protein-centric, or term-centric. For example, the GO term GO:1901184 marks the process "regulation of ERBB signaling pathway" and genes are annotated with this term when they are involved in the process, while KEGG's hsa04012 pathway indexes the relationships between genes participating in the "ERBB Signaling Pathway." In this particular case, the GO term is conceptually the same as the pathway, but without topology. Ontologies aim for greater specificity for each gene while pathways aim for greater resolution on a particular process or function. If similar terms, such as the GO term for ERBB signaling and the KEGG pathway for ERBB signaling, are included in the same analysis, false-discovery correction may be more stringent than is needed. Thus, functions within RITAN to identify, and filter resources base on semi-redundant or highly overlapping concepts, will assist the interpretation of data derived from high-throughput sequencing.

We also distinguish between a discovery *p*-value and an annotation *p*-value. For instance, when calculating differentially expressed genes, a discovery *p*-value is generated and is one of the criteria for calling a gene differentially expressed. False-discovery or other statistical controls should be used at this stage. After discovering differentially expressed genes, RITAN can be a powerful tool to assist interpreting those results. The *p*-values generated in annotation have a different interpretation and role from discovery. Researchers should always be aware of weather they are using a statistical test for discovery (inference) or annotation. Iterating between discovery and annotation should be strictly avoided or explicitly identified—iterating can be a powerful discovery approach, but it can quickly lead to over-fitting. Statistical frameworks for combined feature selection and classification exist and could be adapted to implement an iterative approach in this setting.

We have active plans to expand RITAN's capabilities by adding additional statistical testing methods (*Subramanian et al., 2005*), supporting improved resource-reduction methods, and novel metrics for the statistical significance of network overlaps

(*Ding et al., 2016*; *Zimmermann et al., 2016*). Non-programers could benefit from the capabilities of RITAN. Thus, we have developed a self-contained executable application and made the app available for download on RITAN's GitHub page. We plan to further develop both resources—to improve the core features of RITAN and expand our portable application for use by a broader audience. The features currently within RITAN are highly useful and their implementation in a common framework facilitates ease of use and reproducible research.

## CONCLUSION

Rapid integration of term annotation and network resource fills a niche in being a simple gene-centric knowledge management system—to index geneset and network resources—as well as provide multiple tools for resource integration, assessment, enrichment, and exploration. We believe RITAN will be an enabling tool for the interpretation of high-throughput genomic studies.

## ACKNOWLEDGEMENTS

The authors thank Caroline L. Vitse for her editorial assistance.

### Funding
This work was supported by NIH grant U01AI089859. The funders had no role in study design, data collection and analysis, decision to publish, or preparation of the manuscript.

### Grant Disclosure
The following grant information was disclosed by the authors:
NIH: U01AI089859.

### Competing Interests
The authors declare that they have no competing interests.

### Author Contributions
- Michael T. Zimmermann conceived and designed the experiments, performed the experiments, analyzed the data, contributed reagents/materials/analysis tools, prepared figures and/or tables, authored or reviewed drafts of the paper, approved the final draft.
- Brian Kabat performed the experiments, analyzed the data, contributed reagents/materials/analysis tools, authored or reviewed drafts of the paper, approved the final draft.
- Diane E. Grill analyzed the data, contributed reagents/materials/analysis tools, authored or reviewed drafts of the paper, approved the final draft.
- Richard B. Kennedy authored or reviewed drafts of the paper, approved the final draft, provided feedback on application areas.
- Gregory A. Poland authored or reviewed drafts of the paper, approved the final draft, provided feedback on application areas.

## Data Availability

Data is available at GitHub: https://github.com/MTZimmer/RITAN.

## Supplemental Information

Supplemental information for this article can be found online at http://dx.doi.org/10.7717/peerj.6994#supplemental-information.

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
