# Peer review of "RITAN: rapid integration of term annotation and network resources"

_PeerJ, doi:10.7717/peerj.6994_

## Round 0.1 · original submission · Major Revisions

Dear Dr. Zimmermann and colleagues:

Thanks for submitting your manuscript to PeerJ. I have now received two independent reviews of your work, and as you will see, the reviewers raised some issues about the research. Despite this, these reviewers are optimistic about your work and the potential impact it will have on the bioinformatics research community. Thus, I encourage you to revise your manuscript accordingly, taking into account all of the concerns raised by both reviewers.

While the concerns of the reviewers are relatively minor, this is a major revision to ensure that the original reviewers have a chance to evaluate your responses to their concerns. My overall concern is that RITAN may not be “user-friendly” in a sense that non-computational biologists may struggle with implementation. Provided this, and per reviewer 2’s concerns, please address this directly in your revision.

I look forward to seeing your revision, and thanks again for submitting your work to PeerJ.

Good luck with your revision,

-joe

Reviewer 1 ·

Basic reporting

Figure 1 legend:
Starting from Line 396 “Adding a study design matrix above ...”, you are already talking about Figure 1C, but this is not indicated. Same for Line 403, you wrote “panel B” but seems to be discussing “panel C”.

Experimental design

Vignette needs update. Please also check other parts.
I ran the “Quick Start” code and found that the parameter “resources” should be “term_sources”.
(https://rdrr.io/bioc/RITAN/f/vignettes/enrichment.Rmd).

Validity of the findings

The authors describe a package (RITAN) that can annotate a list of genes with functions using a large collection of gene function definitions (16 term annotations from public databases). The method is similar to existing ones: using hypergeometric test to check if the list of input genes overlap significantly with another group of genes with a defined function. RITAN can estimate similarities between functional-related gene groups to avoid including redundant gene sets. RITAN can also handle user defined functional-related gene groups. These functions will greatly save researchers’ effort when trying to use multiple data sources.
Although many of the functions are already provided by published tools such as DAVID https://david.ncifcrf.gov/, RITAN still has its merit:
The programing interface of RITAN makes it a better choice for researchers with R programing skills. It can also easily generate a nice heatmap to visualize the q-values.

Additional comments

1. Results and discussions are all in one section “Discussion”. PeerJ does offer a “Results” section, so it is better to separate "results" and "discussion".

2. Figure 1 legend is too long. Many of the content belongs to “Results”. Please separate them if possible.

Reviewer 2 ·

Basic reporting

1. Basic reporting:


The authors provided examples to explain how RITAN can facilitate hypothesis generation and exploration in research. However, they do not explain the methodology to implement RITAN, which should be part of this manuscript. Is the methodology of RITAN better than other existing resources?

The figures are not well explained. The authors need to be careful about writing/formatting as well. For example:
In the figure legend for 1B fits for 1C.
There is no explanation of figure 1 in the main text. The authors should simplify the figure legend for figure 1 and explain the figure in detail in the main text (line 135).
In Figure 2, parts of the labels are cut off by the margin. In 2B, the labels are overlapping.
The presentation is really sloppy.

There is no result section, which makes little sense, Instead, the result section is integrated in the discussion...WHY?

Experimental design

The purpose of their tool is to assist researchers with the annotation of their gene sets derived from high-throughput studies. It is not obvious who their user target is. Basic biology researchers are not familiar with R programing. Hence, an additional web interface for users would be much more helpful for a broad audience.

Validity of the findings

As the author mentioned, there are already many resources that standardize and codify different databases for gene function from different perspectives. The authors failed to address what the limitations of these resources are and how RITAN performed better than those. A comparison is important.

Additional comments

It is not user-friendly and cannot be used without using R. Hence, impact is limited. Also, the presentation is sloppy (e.g. Figure 2B overlapping text; Figure 3A, basal versus basel, etc.)

---

## Round 0.2 · accepted · Accept

Dear Dr. Zimmermann and colleagues:

Thanks for revising your manuscript based on the minor concerns raised by the reviewers. I now believe that your manuscript is suitable for publication. Congratulations! I look forward to seeing this work in print, and I anticipate it being an important resource for research communities studying genomics and bioinformatics. Thanks again for choosing PeerJ to publish such important work.

Best,

-joe

# Reviewer 1 ·

Basic reporting

All concerns have been addressed in this version.

Experimental design

All concerns have been addressed in this version.

Validity of the findings

All concerns have been addressed in this version.

Additional comments

All concerns have been addressed in this version.

Reviewer 2 ·

Basic reporting

The authors have made proper corrections to address our concerns.
One minor issue: Figure 2A, the color scale is missing when the file is visualizing using Preview. The figure looks fine with Adobe reader though.

Experimental design

The authors have separated the result and discussion section that is much clearer now.

Validity of the findings

None